# Effectiveness of postdischarge interventions for reducing the severity of chronic pain after total knee replacement: systematic review of randomised controlled trials

Vikki Wylde,[1,2] Jane Dennis,[1] Rachael Gooberman-Hill,[1,2] Andrew David Beswick[1]

[1]Musculoskeletal Research Unit, Translational Health Sciences, Bristol Medical School, University of Bristol, Bristol, UK
[2]National Institute for Health Research Bristol Biomedical Research Centre, University Hospitals Bristol NHS Foundation Trust and University of Bristol, Bristol, UK

**Correspondence to**
Dr Vikki Wylde;
v.wylde@bristol.ac.uk

## ABSTRACT

**Objective** Approximately 20% of patients experience chronic pain after total knee replacement (TKR). The aim of this systematic review was to evaluate the effectiveness of postdischarge interventions commenced in the first 3 months after surgery in reducing the severity of chronic pain after TKR.

**Design** The protocol for this systematic review was registered on PROSPERO (registration number: CRD42017041382). MEDLINE, Embase, CINAHL, PsycINFO and The Cochrane Library were searched from inception to November 2016. Randomised controlled trials of postdischarge intervention which commenced in the first 3 months after TKR surgery were included. The primary outcome of the review was self-reported pain severity at 12 months or longer after TKR. Risk of bias was assessed using the Cochrane risk-of-bias tool.

**Results** Seventeen trials with data from 2485 randomised participants were included. The majority of trials evaluated physiotherapy interventions (n=13); other interventions included nurse-led interventions (n=2), neuromuscular electrical stimulation (n=1) and a multidisciplinary intervention (n=1). Opportunities for meta-analysis were limited by heterogeneity. No study found a difference in long-term pain severity between trial arms, with the exception of one trial which found home-based functional exercises aimed at managing kinesiophobia resulted in lower pain severity scores at 12 months postoperatively compared with advice to stay active.

**Conclusion** This systematic review and narrative synthesis found no evidence that one type of physiotherapy intervention is more effective than another at reducing the severity of chronic pain after TKR. Further research is needed to evaluate non-physiotherapy interventions, including the provision of care as part of a stratified and multidisciplinary care package.

**PROSPERO registration number** CRD42017041382.

## Strengths and limitations of this study

► This is the first systematic review to evaluate the effectiveness of postdischarge interventions delivered in the first 3 months after surgery in reducing the severity of chronic pain after total knee replacement.
► Synthesis of adverse events data was not possible because assessment and reporting were variable and often poor.
► We did not include studies that used a composite pain and function measure to assess pain outcome.

chronic pain in the months and years after TKR. Chronic postsurgical pain is defined as pain that is present or increases in intensity at ≥3 months after surgery.[1] In representative populations, unfavourable long-term pain outcomes have been reported by 10%–34% of patients with TKR.[2] Patients with bothersome pain at ≥3 months after surgery are disappointed with their outcome.[3 4] Given the prevalence and impact of chronic pain, it is important to evaluate interventions that may optimise patients' outcomes after TKR.

During the hospital stay after TKR, rehabilitation focuses on regaining range of motion and improving mobility. After discharge, rehabilitation aims to enhance recovery, through supporting a person to regain function and quality of life, optimising pain relief and reintegration into social and personal environments.[5] While physiotherapy often focusses on functional health, another key outcome is the prevention of long-term pain.[6] Postoperative physiotherapy may be combined with other interventions to provide comprehensive, multidisciplinary and holistic rehabilitation.[7] A key step to improving patients' outcomes after TKR is to evaluate if early postoperative rehabilitation interventions

## INTRODUCTION

Total knee replacement (TKR) is a common operation to provide pain relief, predominately due to osteoarthritis. Despite good outcomes for many, some patients report

can reduce the severity of chronic pain after TKR. Chronic pain is difficult to treat once established,[8] and therefore, it is important to evaluate the effectiveness of early postoperative interventions in reducing the severity of chronic pain.

The aim of this systematic review was to evaluate the effectiveness of postdischarge interventions delivered in the first 3 months after surgery for reducing the severity of chronic pain after TKR.

## METHODS

The review was registered on the international prospective register of systematic reviews (PROSPERO) on 17 January 2017 (registration number: CRD42017041382). The review was conducted following guidance from the Cochrane Handbook[9] and reported in accordance with Preferred Reporting Items for Systematic Reviews and Meta-Analyses guidelines[10] (see online supplementary appendix 1).

### Eligibility criteria

Studies were eligible for inclusion in the review if they met the following criteria:

► Population: Adults discharged from hospital after primary TKR predominantly for osteoarthritis.
► Intervention: Any postdischarge intervention which commenced in the first 3 months after TKR surgery.
► Control: Any, including no intervention, usual care, placebo or an alternative intervention.
► Outcomes: The primary outcome was pain severity at 12 months or longer after TKR, as patient-reported levels of pain plateau by this time point.[11 12] Pain severity could be assessed using a patient-reported joint-specific pain measure (eg, Western Ontario and McMaster Universities Osteoarthritis Index (WOMAC) or Knee Injury and Osteoarthritis Outcome Score (KOOS) pain domains), a quality-of-life measure (eg, Short-Form (SF-36 or SF-12) or a Visual Analogue Scale (VAS). The secondary outcome was adverse events.
► Study type: randomised controlled trials (RCTs).

### Information sources and searches

MEDLINE, Embase, CINAHL, PsycINFO and The Cochrane Library were searched from inception to 15 November 2016 (see online supplementary appendix 2). No language restrictions were applied and relevant non-English articles were translated and included if appropriate. Studies reported only as abstracts or that were unobtainable as full text copies using interlibrary loans or email contact with authors were excluded. Citations of key reviews and studies were checked in Institute for Scientific Information (ISI) Web of Science.

### Screening

Records identified by searches were imported into Endnote X7 (Thomson Reuters) and duplicates removed. From the searches, an Endnote database of all RCTs and systematic reviews in TKR was established. Within this database, interventions conducted during the postoperative period were identified. An initial screen for potential eligibility was undertaken by one reviewer (ADB) to exclude articles that were clearly not relevant. Subsequently, abstracts and full-text articles were screened independently by two reviewers (VW and ADB or JD). Results of screening were compared, and any discrepancies were resolved through further review of the full-text articles and discussion between reviewers. Reasons for exclusion were recorded.

### Data extraction

Data from studies that met the eligibility criteria were extracted onto a standardised pro forma by one reviewer (VW). Data extraction was checked against source articles by a second reviewer (JD). Extracted data comprised: country, date, participant characteristics, selection criteria; intervention and control treatment; follow-up intervals; losses to follow-up; primary outcome; outcome data for pain; adverse events (any untoward medical occurrence in a clinical study participant regardless of the causal relationship with the study treatment) and information for risk of bias assessment. Any disagreements between reviewers were discussed with a third reviewer (ADB) and consensus reached.

A single email was sent to authors of studies with an appropriate follow-up period but no pain outcome to enquire if an appropriate outcome was available. If a combined pain and function outcome was reported, such as the Oxford Knee Score (OKS) or the total WOMAC score, separate pain subscale data were requested. Authors were contacted when necessary for clarification purposes or to request unpublished relevant data. If a study reported data that were combined for knee and hip replacement patients, then disaggregated data for patients with TKR were requested. If this was not available, then the study was excluded.

### Risk of bias assessment

Potential sources of bias were assessed using the Cochrane risk of bias tool.[9] At the protocol stage, analysis was planned which included all studies, with sensitivity analyses conducted to exclude studies judged to be at high risk of bias.

### Strategy for data synthesis

At the protocol stage, meta-analysis using RevMan 5[13] was planned if two or more studies were identified with similar interventions and comparator groups and appropriate outcome data. If continuous pain outcomes were measured differently across studies, overall standardised mean differences and 95% CIs would be calculated and presented alongside measures of heterogeneity ($I^2$). Where possible, subgroup analyses were planned to explore the effectiveness of different intervention content and intensity, and different comparator interventions.

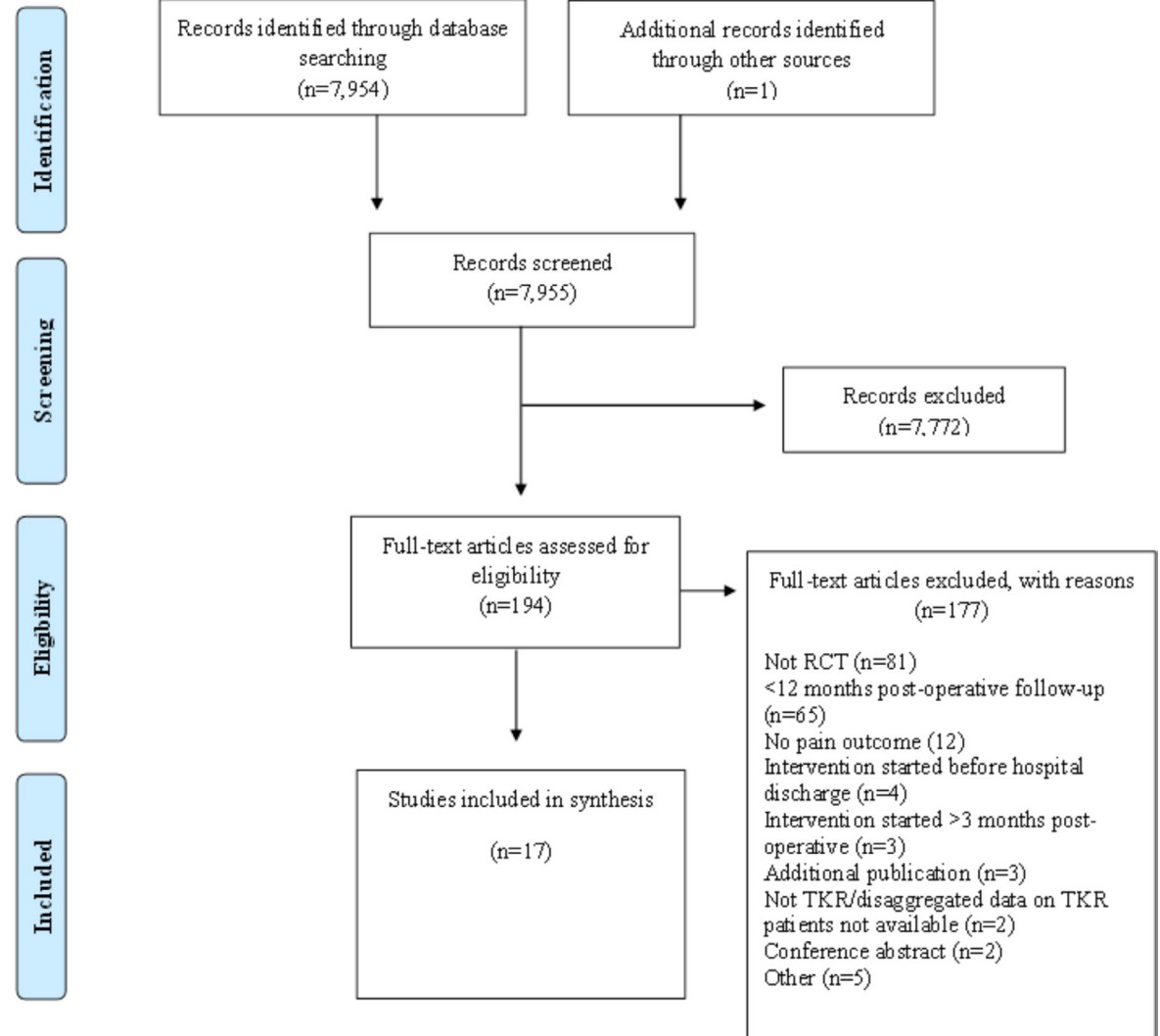

**Figure 1** Systematic review flow diagram. RCT, randomised controlled trial; TKR, total knee replacement.

Opportunities for pooling outcome data in meta-analysis were limited by heterogeneity. This included the content, duration and intensity of the treatments in both the intervention and comparison groups. For example, a number of the trials were pragmatic with the control group receiving 'usual care', which varied considerably between studies. Therefore, a narrative synthesis was performed.

### RESULTS

Searches identified 7954 articles. After detailed evaluation of full-text articles, 17 studies with 2485 randomised participants were included[14–30] (figure 1). Two included studies were published after the search dates, but were identified from protocols published within the search dates.[15 17]

### Study characteristics

Table 1 provides an overview of study characteristics. Included studies were from Australia (n=3), Canada (n=2), Finland (n=2), Germany (n=2), UK (n=2), China (n=1), Denmark (n=1), Italy (n=1), Norway (n=1),

Turkey (n=1) and USA (n=1). The number of centres was reported for 15 studies: eight studies were conducted in a single centre, three studies were conducted in two centres and four studies were conducted in ≥4 centres. Sample sizes ranged from 34 to 422 participants, with a median of 117. All studies had two arms, with the exception of one three-arm trial.[20] Three studies were described as pilot or feasibility studies.[16 18 24]

### Study quality

Risk of bias assessments for individual studies are displayed in figure 2. All studies were at high risk of bias for blinding of participants and pain outcome assessment due to the nature of the intervention and the self-reporting of pain. Five studies were at high risk of bias due to incomplete outcome data and one due to selective outcome reporting.

### Outcomes assessment

The primary outcome was specified for 13 trials; this was function in eight trials, a composite of pain and function in four trials and pain in one trial (see online

**Table 1** Overview of study characteristics

| Publication location date of study number of centres | Randomised mean age % female | Intervention treatment | Control treatment | Pain assessment adherence to treatment Losses to follow-up |
|---|---|---|---|---|
| Bruun-Olsen et al[14] Norway 2008–2010 Two centres | n=57 (29:28) 68:69 years 62:50% | Group-based physiotherapist-led walking skills programme (2–6 patients per group). Commenced 6 weeks after surgery. Twelve sessions over 6–8 weeks. | 1:1 usual physiotherapy care consisting of 12 individual physiotherapy sessions. Commenced 6 weeks after surgery. Twice-weekly sessions until 12–14 weeks after surgery. | KOOS pain scale 28/29 completed intervention. 28/28 received control treatment. 6 (2:4) lost to follow-up. |
| Buhagiar et al[15] Australia 2012–2015 Two centres | n=165 (81:84) 67:67 years 69:68% | Inpatient rehabilitation at rehabilitation facility with twice daily supervised sessions of 1:1 and group-based exercises. Commenced after hospital discharge for 10 days. Home exercise programme after discharge from rehabilitation facility. | Home exercise programme comprising 2–3 group-based outpatient sessions to practice and progress exercises. Commenced 2 weeks after surgery. | KOOS pain scale 72/81 adhered to intervention. 74/84 adhered to control treatment. 6 (2:4) lost to follow-up. |
| Buker et al*[30] Turkey 2009–2011 One centre | n=34 (18:16) 64:68 years 89:94% | 20 sessions of supervised physiotherapy and rehabilitation including range of motion and strengthening exercises, application of heat and TENS application. Five days a week for 4 weeks. | Home exercises including range of motion and strengthening exercise for 1 hour per day. Five days a week for 4 weeks. | Pain VAS Adherence not reported. Losses to follow-up not reported. |
| Chen et al[16] China 2013–2014 One centre | n=202 (101:101) 66:67 years 63:67% | Structured telephone follow-up by nurse at 1, 3 and 6 weeks after hospital discharge to improve adherence to home exercise routine. | No telephone follow-up | Pain VAS Adherence not reported. 15 (7:8) lost to follow-up. |
| Fransen et al[17] Australia 2009–2012 12 centres | n=422 (212:210) 64:65 years 54:52% | Group-based circuit exercise classes supervised by physiotherapist. Up to six patients per class. Commenced 6 weeks after surgery. Twice-weekly sessions for at least 8 weeks. | Usual physiotherapy care. Twenty-two per cent of participants reported six or more occasions of physiotherapy during the 6–12-week period after TKR. | WOMAC pain scale 140/212 participants attended≥12 classes 210/210 received control treatment 74 (43:31) lost to follow-up |
| Frost et al[18] UK 1995–1996 Not reported | n=47 (23:24) 72:71 years 48:50% | Home-based functional exercise. Commenced following discharge from hospital. Duration not reported. | Home-based traditional exercise. | OKS item (pain on walking) Adherence not reported. 20 (7:13) lost to follow-up. |
| Kauppila et al[19]† Finland 2002–2005 One centre | n=86 (44:42) 71:71 years 76:79% | Group-based multidisciplinary rehabilitation programme. Up to eight patients per group. Commenced 2–4 months after surgery for 10 days. | Usual physiotherapy care. Supervised exercise programme at 2-month outpatient visit, with provision of further rehabilitation based on needs assessment. | WOMAC pain scale 44/44 attended intervention. 42/42 received control treatment. 11 (8:3) lost to follow-up. |
| Ko et al[20] Australia 2008–2010 Three-arm trial Four centres | n=249 (85:84:80) 67:68:67 years 68:60:61% | 1:1 physiotherapy with home-based sessions. Commenced 2 weeks after surgery. Twice-weekly 1:1 and home-based sessions over 6 weeks. | (1) Group-based circuit classes supervised by physiotherapist with home-based sessions. Up to eight patients per class. Commenced 2 weeks after surgery. Twice-weekly group and home-based sessions over 6 weeks. (2) Monitored home programme, two 1:1 physiotherapy sessions and one telephone follow-up call. Commenced 2 weeks after surgery. Four sessions per week for 6 weeks. | WOMAC pain scale 80% participants attended 9 or more 1:1 sessions, 77% attended nine or more group sessions, 83% attended both sessions in monitored home programme group. 16 (7:3:6) lost to follow-up. |

Continued

**Table 1** Continued

| Publication location date of study number of centres | Randomised mean age % female | Intervention treatment | Control treatment | Pain assessment adherence to treatment Losses to follow-up |
|---|---|---|---|---|
| Kramer et al[21] Canada Not reported Not reported | n=160 (80:80) 68:69 years 59:55% | 1:1 clinic-based rehabilitation programme with home exercise programme. Commenced 2 weeks after surgery. Up to two sessions a week for 10 weeks. | Home-based rehabilitation, monitored by telephone calls from physiotherapist. Commenced 2 weeks after surgery. At least one telephone call in weeks 2–6 and 1 call in weeks 7–12. | WOMAC pain scale 76/80 received intervention. 78/80 received control treatment. 26 (15:22) lost to follow-up. |
| Liebs et al*[22] Germany 2005–2006 Five centres | n=159 (85:74) 70:70 years 73:70% | Ergometer cycling supervised by physiotherapist. Commenced 2 weeks after surgery. Three sessions a week for at least 3 weeks. | No ergometer cycling. | WOMAC pain scale Adherence not reported. 33 (15:18) lost to follow-up. |
| Liebs et al*[23] Germany 2003–2004 Two centres | n=185 (87:98) 69:71 years 70:73% | Early aquatic therapy. Commenced 6 days after surgery. Three times a week up to fifth week postoperative. | Late aquatic therapy. Commenced 14 days after surgery. Three times a week up to fifth week postoperative. | WOMAC pain scale Adherence not reported. 41 (18:23) lost to follow-up. |
| Minns Lowe et al[24] UK 2006–2008 One centre | n=107 (56:51) 68:71 years 57:59% | Home-based functional rehabilitation with two visits from physiotherapist at 2 weeks and 6–8 weeks after hospital discharge. Twice daily exercises for at least 3 months. | Usual physiotherapy care involving provision of an exercise booklet, with outpatient physiotherapy on a needs-only basis. No additional home visits. | KOOS pain scale 46/56 patients received two visits. 47/51 received control treatment. 9 (7:2) lost to follow-up. |
| Moffet et al[25] Canada 1997–1999 Five centres | n=77 (38:39) 67:69 years 63:56% | Functional rehabilitation programme with individualised home exercises. Commenced at 2 months after surgery. Twelve supervised sessions over 6–8 weeks. | Usual physiotherapy care, which included supervised home rehabilitation visits for 26% of patients. | WOMAC pain scale 38/38 participated in 12 sessions. 39/39 received control treatment. 8 (0:8) lost to follow-up. |
| Monticone et al[26] Italy 2010–2013 One centre | n=110 (55:55) 67:68 years 65:62% | Home-based functional exercises aimed at managing kinesiophobia, with monthly phone calls to encourage adherence. Commenced after hospital discharge. Twice-weekly sessions for 6 months. | No physiotherapy, advice to stay active. | KOOS pain scale Adherence not reported. 0 lost to follow-up. |
| Petterson et al[27] USA 200–2005 One centre | n=200 (100:100) 65:65 years 47:45% | Combined neuromuscular electric stimulation (NMES) and volitional strength training programme. Commenced 3–4 weeks after surgery. Two or three sessions a week for 6 weeks. | Volitional strength training programme without NMES. | KOS ADL item (effect of pain on function) 84/100 completed intervention. 97/100 completed control treatment. 51 (32:19) lost to follow-up. |
| Szots et al[28] Denmark 2013 One centre | n=117 (59:58) 67:68 years 61:67% | Two nurse-led structured telephone follow-up calls. Telephone calls at 4 days and 14 days after hospital discharge. | No telephone follow-up. | WOMAC pain scale 54/59 patients had both telephone follow-up calls. 54/58 received control treatment. 9 (5:4) lost to follow-up. |
| Vuorenmaa et al[29]‡ Finland 2008–2010 One centre | n=108 (53:55) 69:69 years 57:65% | Delayed monitored home exercises, with guidance from physiotherapist at 2, 3 and 6 months postoperative. Commenced at 2 months after surgery for 12 months. | Usual care, which involved no additional guidance from 2 months postoperative. | WOMAC pain scale Seventy-two per cent of patients performed the training sessions at least twice per week in the first 6 months. 53/53 received control treatment. 4 (2:2) lost to follow-up. |

*24-month follow-up also conducted but data from 12-month follow-up included in table to be consistent with follow-up period of other studies.
†Pain-specific outcome data were provided by the authors for a previous review[6] and was used again in this review.
‡Follow-up at 14 months postoperatively.
KOOS, Knee Injury and Osteoarthritis Outcome Score; KOS ADL, Knee Outcome Survey - Activities of Daily Living; OKS, Oxford Knee Score; TENS, Transcutaneous Electrical Nerve Stimulation; VAS, Visual Analogue Scale; WOMAC, Western Ontario and McMaster Universities Osteoarthritis Index.

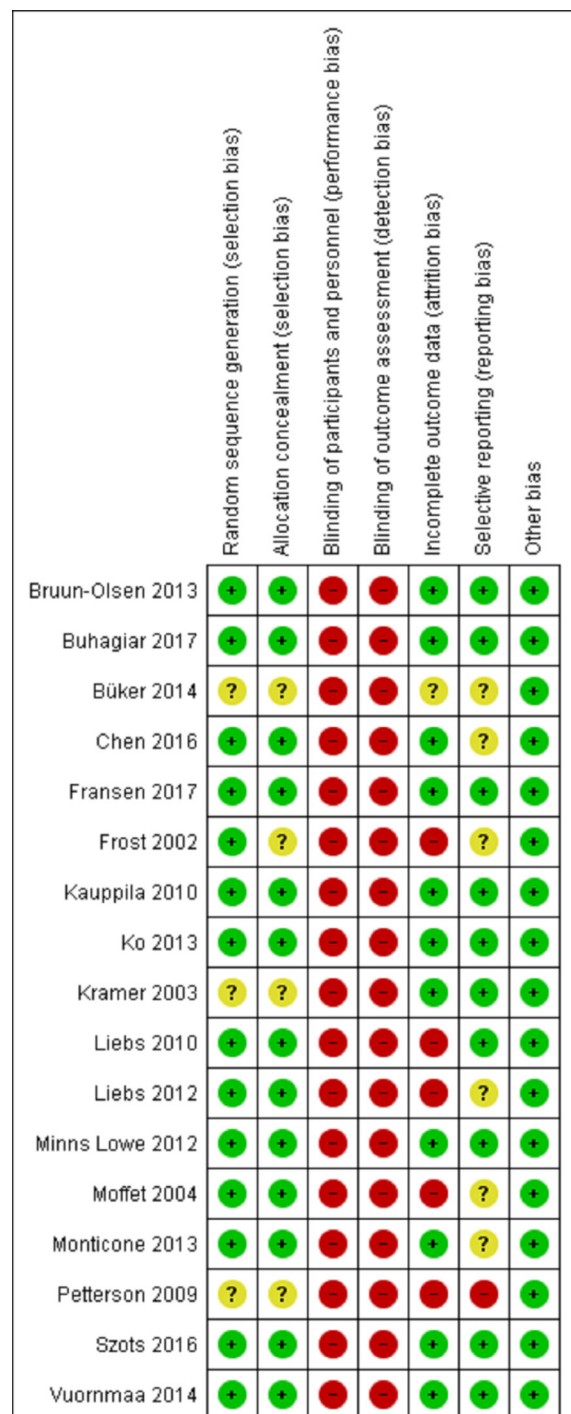

**Figure 2** Risk of bias assessment for individual studies.

supplementary appendix 3). Pain severity was most commonly assessed using the WOMAC pain scale (n=9); other tools included the KOOS pain scale (n=4), VAS (n=2) and single items from the OKS (n=1) and KOS ADL (n=1). Adverse events were poorly described and reported in the majority of studies and therefore pooling of harms data was not possible. A summary of adverse events findings is presented in online supplementary appendix 4. Pain was assessed at 12 months after TKR in 16 studies and at 14 months in one study. A summary of

results from the individual studies is provided in online supplementary appendix 3.

### Interventions
The majority of studies evaluated physiotherapy interventions (n=13); other interventions evaluated included nurse-led interventions (n=2), neuromuscular electrical stimulation (NMES) (n=1) and a multidisciplinary intervention (n=1).

### Physiotherapy interventions
Thirteen studies with 1880 randomised patients evaluated the effectiveness of postdischarge physiotherapy interventions. All interventions started within 2 months of surgery, with the majority commencing within 2 weeks of surgery. In addition to all studies being at risk of bias due to issues with blinding, risk of bias due to incomplete outcome data was evident for four studies.[18 22 23 25] Seven studies compared physiotherapy interventions with usual care or minimal care; interventions included a walking skills programme,[14] group-based circuit exercise classes,[17] ergometer cycling,[22] home-based functional rehabilitation,[24] clinic-based functional rehabilitation,[25] home-based functional exercises aimed at managing kinesiophobia[26] and delayed monitored home exercises.[29] Five studies compare two forms of treatment including inpatient rehabilitation compared with home exercise,[15] home-based functional exercise and home-based traditional exercise,[18] 1:1 physiotherapy and home-based rehabilitation,[21] supervised and home-based physiotherapy,[30] early aquatic therapy and late aquatic therapy[23] and a three-arm trial comparing 1:1 physiotherapy, group-based circuit classes and a monitored home exercise programme.[20] Of the 13 studies, only one trial reported a difference in pain severity between groups; patients randomised to 6 months of home-based exercises aimed at managing kinesiophobia had lower pain severity scores at 12 months postoperative compared with patients who received general advice to stay active.[26]

### Nurse-led interventions
Two studies with 319 randomised patients reported evaluation of a nurse-led intervention compared with no care or usual care. Except for issues relating to blinding, both studies were at low risk of bias. Both studies evaluated nurse-led structured telephone follow-up; one aimed to improve adherence to home exercise[16] and the other to provide information regarding well-being, integrity, prophylaxis, safety and other issues relevant to patients after TKR.[28] Pain outcome data (mean and SD) were not available for latter study and, therefore, meta-analysis was not possible. Neither study found a difference in pain severity scores at 12 months postoperative between the intervention and control group.

### Other interventions
Two studies reported evaluations of other interventions. Except for issues relating to blinding, both studies were at low risk of bias. One trial involving 86 patients compared

a group-based multidisciplinary programme with usual care.[19] This 10-day programme involved physiotherapy, Nordic walking, relaxation strategies and sessions with a psychologist, social worker, nutritionist and orthopaedic surgeon. Another trial with 200 patients, at high risk of bias due to incomplete outcome data and selective outcome reporting, evaluated a combined NMES and volitional strength training programme compared with volitional strength training programme without NMES.[27] Both studies found no difference in pain severity scores at 12 months postoperative between the intervention and control group.

## Ongoing research

In searches of databases and citation searches on ISI Web of Science, we identified a number of published RCT protocols that are evaluating postdischarge interventions with a pain severity outcome at ≥12 months after TKR. Interventions being evaluated include a digital activity coaching system for home exercise,[31] Wii-enhanced rehabilitation,[32] group-based outpatient physiotherapy with an individualised element,[33] multicomponent rehabilitation for patients at risk of a poor outcome[34] and physiotherapy for patients performing poorly at 6 weeks after TKR.[35] Some of these studies are now finished and findings are likely to be reported imminently.

## DISCUSSION

This systematic review aimed to evaluate the effectiveness of postdischarge interventions delivered in the first 3 months after surgery for reducing the severity of chronic pain after TKR. Interventions that predominately comprise physiotherapy have been evaluated in RCTs. In most studies, the control group received some form of physiotherapy care and, therefore, the aim of the trials was to compare the effectiveness of different types of physiotherapy, rather than to compare the effectiveness of physiotherapy to no care. A narrative synthesis of the evidence suggests that no physiotherapy intervention appears to be more effective than another at reducing the severity of chronic pain after TKR. However, findings from the trial of a 6-month home-based functional exercise programme aimed at managing kinesiophobia[26] compared with advice to stay active were encouraging and warrant further evaluation. Few studies have been conducted to evaluate the effectiveness of non-physiotherapy interventions at reducing chronic pain after TKR, and further research is needed.

There are a number of strengths and limitations to this systematic review. The main outcome of interest in this review was pain severity at ≥12 months after TKR. Although the primary outcome of many of the included studies was function, pain severity was an important secondary outcome in these studies. Studies that used a composite pain and function measure to assess outcome, for example, the OKS or WOMAC, were excluded if authors were unable to provide pain subscales scores. Although this reduced the number of studies eligible for inclusion, this approach was taken because pain and function are distinct outcome domains,

with different predictors and recovery trajectories.[36 37] The secondary outcome of this review was adverse events, to allow the synthesis of harms data. However, synthesis was not possible because assessment and reporting of adverse events was variable and often poor. The quality of adverse events reporting is a common issue in surgical trials,[38] and evidence-based recommendations are needed to promote standardisation, improve quality and reduce heterogeneity of adverse events reporting in orthopaedic studies. A potential limitation of the included studies was that they were all at high risk of bias due to the lack of participant blinding for self-report pain. However, blinding of participants is rarely possible in RCTs of this nature. Also, it would be expected that the risk would arise from participants in the intervention group reporting less pain, which may potentially be an issue with shorter-term outcomes, but this was not evident from the longer-term follow-up of the studies included in this review.

This systematic review took a broad approach by evaluating the effectiveness of any type of postdischarge intervention that aimed to reduce the severity of chronic pain after TKR. Interventions that span the postoperative period may be delivered as part of a comprehensive perioperative package of care, and these would not have been identified in this review; however, evaluations of the effectiveness of preoperative and perioperative interventions for reducing chronic pain severity are being conducted separately (CRD42017041382). Previous systematic reviews of interventions to improve long-term outcomes after TKR have been conducted, but these have evaluated preoperative interventions or have been narrower in focus. Systematic reviews of preoperative interventions have found that exercise and education have a limited effect on improving long-term pain and function after TKR.[39–43] Previous systematic reviews of postdischarge interventions have focused on physiotherapy, finding some evidence of short-term benefit but a lack of evidence to draw conclusions about long-term benefit.[6 44 45] One systematic review has evaluated interventions for the management of chronic pain after TKR, identifying only a single RCT of botulinum toxin A injections.[46] Our systematic review adds to this literature by providing evidence that no specific type of postdischarge physiotherapy intervention appears to be more effective than another at reducing the severity of chronic pain after TKR, although the positive impact of a home-based programme aimed at managing kinesiophobia compared with advice to stay active warrants further investigation.

The aim of this review was to evaluate the effectiveness of postdischarge interventions at reducing chronic pain severity after TKR. However, the primary aim of most trials included in the review was to improve functional ability after TKR. Only one trial had a primary outcome of pain severity,[29] although a number of other trials assessed their primary outcome with a composite measure of pain and function.[17 20 24 26] However, pain severity was assessed as a secondary outcome in these trials and, therefore, it was expected that the intervention may reduce long-term

pain. All but one study found that the intervention did not provide any benefit on long-term pain severity compared with the control group. However, the treatment received in the control group, particularly in the physiotherapy trials, varied considerably between studies, including a different form or intensity of physiotherapy, provision of physiotherapy based on a needs assessment, delayed treatment or no treatment. Therefore, it is not appropriate to draw conclusions on the effectiveness of any particular type of physiotherapy intervention based on the findings of this review. However, the evidence does suggest that no type of physiotherapy intervention is more effective than another at reducing the severity of chronic pain after TKR. An important finding of this review is that only four trials have been conducted which have evaluated non-physiotherapy interventions, highlighting the need for more research in this field. In particular, further research with pain severity as the pain outcome is needed to ensure that RCTs are adequately powered to evaluate the effectiveness of postdischarge interventions on reducing chronic pain severity.

There are important considerations in the design and delivery of future postdischarge interventions that warrant further discussion. All the interventions included in this review were uniformly delivered to all patients, rather than just those patients who may be at most risk of poor outcome. Only 20% of patients will develop chronic pain after TKR[2] and delivering physiotherapy to all patients may reduce the ability to detect clinical benefit in terms of pain severity. In the future, interventions might be more effective if they include processes to identify patients at high risk of chronic pain and provide these patients with intensive and comprehensive interventions that have been specifically designed to reduce their risk of developing chronic pain. However, identifying high risk patients is challenging; preoperative models to identify patients at risk of a poor outcome have low predictive power,[36 37] and the evidence for postoperative risk factors is limited.[47] However, research is currently ongoing to evaluate whether providing a rehabilitation programme for patients at risk of a poor outcome[34] or are 'functioning poorly' at 6 weeks after TKR[35] can improve longer-term outcomes.

Preventing chronic pain after TKR is challenging because of the complexity of this pain condition. Although chronic pain after TKR is not yet fully understood, the aetiology of this pain is multifactorial, including surgical factors, complex regional pain syndrome, pain sensitisation, neuropathic pain and psychosocial factors.[48–53] Therefore, an intervention comprising a single treatment modality may be insufficient to address and reduce the causes of pain for all patients. As with the treatment of other chronic pain conditions, this highlights the importance of focused, individualised and multidisciplinary treatment.[8 54] Such an approach is being evaluated in an ongoing RCT of a care pathway for patients with chronic pain after TKR (ISRCTN92545361). Therefore, further research is needed to evaluate the effectiveness of providing stratified and multidisciplinary care packages for preventing chronic pain after TKR.

In conclusion, the finding from this systematic review and narrative synthesis is that there is no evidence that one type of physiotherapy intervention is more effective than another at reducing the severity of chronic pain after TKR. Further research is needed to evaluate non-physiotherapy interventions, including the provision of care as part of stratified and multidisciplinary care package.

**Acknowledgements** The authors would like to thank all the study authors who took the time to reply to our requests for further clarification or additional data.

**Contributors** All authors contributed to the concept and design of the study. ADB, JD and VW contributed to the acquisition and analysis of data. VW drafted the article and ADB, JD and RG-H revised it critically for important intellectual content. VW and ADB take responsibility for the integrity of the work as a whole, from inception to finished article.

**Funding** Research (NIHR) under its Programme Grants for Applied Research programme (RP-PG-0613-20001). This study was supported by the NIHR Biomedical Research Centre at the University Hospitals Bristol NHS Foundation Trust and the University of Bristol.

**Disclaimer** The views expressed are those of the authors and not necessarily those of the NHS, the NIHR or the Department of Health. The funder had no involvement in the study design, data collection, data analysis, interpretation of data or writing of the manuscript.

**Competing interests** None declared.

**Patient consent** Not required.

**Provenance and peer review** Not commissioned; externally peer reviewed.

**Data sharing statement** No additional data are available.

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
