## [Reviewer comments · BMJ Open]

ARTICLE DETAILS

TITLE (PROVISIONAL)	The effectiveness of post-discharge intervention for reducing the severity of chronic pain after total knee replacement: Systematic review of randomised controlled trials
AUTHORS	Wylde, Vikki; Dennis, Jane; Goberman-Hill, Rachael; Beswick, Andrew

VERSION 1 – REVIEW

REVIEWER	Stephen Gill School of Medicine, Deakin University, Australia and Barwon Centre for Orthopaedic Research and Education, St John of God Hospital Geelong, Australia
REVIEW RETURNED	07-Nov-2017

GENERAL COMMENTS	Thank you for the opportunity to review the study. Overall, the manuscript is well written and easy to read. The method is sound, though some minor clarifications would be useful. The results are clearly presented. My main concern is regarding the interpretation of the results and the subsequent conclusion, which I believe need to be reconsidered. Abstract 'Objective' states 'delivered in the first three months'. 'Design' states 'commenced in first three months': there is a subtle yet important difference. Suggest use the term commenced. Method Screening: "full-text articles were screened independently by two reviewers (VW and ADB or JD) and reasons for exclusion recorded." Were results of screening compared, if so how, and were there any discrepancies? Data extraction: define "serious adverse event" and perhaps indicate how this compares to a non-serious adverse event. Appendix 3 is noted – are these only 'serious' adverse events? Figure 1: why exclude interventions that started before hospital discharge and exclude interventions that started > 3 months after the operation (accepting that interventions will need to be complete to 12 months)? Results The meta-analysis included 6 studies. Five studies compared the 'intervention' to 'usual physiotherapy care'. (Table 4 indicates 'usual physiotherapy care', but this is not defined or described; doing so would assist the reader to understand
---

	the comparison intervention). Regarding Figure 2, none of these five studies found a difference between the groups. The remaining study compared exercises with no physio/advice to stay active – this study found a difference between the groups. The conclusion “that post-discharge physiotherapy interventions appear not [to] be effective at reducing the severity of chronic pain after TKR” is not supported by the results, given that 5 of 6 studies compared two types of physio. A more reasonable conclusion from these five studies is that there is no evidence that one type of physiotherapy produces different effects on pain 12 months following the operation compared to a different type of physio/exercise. The conclusion from the remaining study, is that exercise is more effective than advice to stay active. Given the fundamental difference between Monticone’s study and the remaining five, the study’s inclusion in the meta-analysis with the remaining five studies is questionable. Discussion “Only 20% of patients will develop chronic pain after TKR [2] and delivering physiotherapy to all patients may reduce the ability to detect clinical benefit in terms of pain severity. This suggests a need to identify patients at high risk of chronic pain and provide these patients with intensive and comprehensive interventions that have been specifically designed to reduce their risk of developing chronic pain.” Please provide evidence for this suggestion.
--	--

REVIEWER	David Hamilton University of Edinburgh, UK
REVIEW RETURNED	22-Nov-2017

GENERAL COMMENTS	Many thanks for inviting me to review this paper which evaluates the effectiveness of post-discharge TKA interventions in reducing the severity of chronic pain through systematic review. I think this is a nice study that has been performed technically well, as would be expected from this group. I have no concerns with the analysis. I do have a couple of general concerns though. These may simply relate to the wording and discussion as opposed to any flaw in methodology. 1. The study aim was to evaluate the effectiveness of interventions at reducing chronic pain severity, but I don't see much reference to pain severity in the manuscript. I may have misread this, but the entry criteria seems to be a report of pain outcomes by intervention group - and chronic pain defined as pain levels that are equivalent to (or greater than) baseline values? Are you measuring pain severity here, or rather the number of patients with chronic pain in the cohort? 2. Following on from that, the definition of chronic pain is a little vague here - as there seems to be no note or threshold as to intensity? Is this patients with high levels of chronic pain at 12 months - that I would want to refer for chronic pain interventions and/or analgesia review, or is that patients that have not changed their reported pain intensity from pre-op to post-op. If it is the latter, we need to be a little cautious as to the interpretation of the term 'chronic pain'.
--

	3. I wonder as to the rehabilitation goals of the included studies. Specifically, I wonder whether the included studies were aiming to reduce levels of chronic pain or whether they were generally aiming to enhance outcomes. Though clearly similar intentions at face value, I would agree that these are not the same thing. It is interesting that the study that did seem to influence pain outcomes was the one that addressed kinesiophobia. The others that did not influence pain scores seem to be generic exercise based interventions - I would not necessarily expect exercise interventions to address pain. On balance, I think that what has been looked at here are the randomised interventions delivered in the post-discharge period that aim to improve outcomes following TKA (and that record pain levels). I think the authors show a lack of efficacy as to these in reducing pain report at 12 months in relation to the internal control of the study. I agree with the conclusions that uniformly applied intervention don't seem to be particularly effective at influencing outcomes - but i'm not entirely sure the authors can comment on chronic pain severity? Perhaps i'm mis-reading things, but I think there needs to be a bit more context given as to what is actually being measured here.
--	--

VERSION 1 – AUTHOR RESPONSE

REVIEWER 1: STEPHEN GILL

Thank you for the opportunity to review the study. Overall, the manuscript is well written and easy to read. The method is sound, though some minor clarifications would be useful. The results are clearly presented. My main concern is regarding the interpretation of the results and the subsequent conclusion, which I believe need to be reconsidered.

Comment 1

Abstract: 'Objective' states 'delivered in the first three months'. 'Design' states 'commenced in first three months': there is a subtle yet important difference. Suggest use the term commenced.

Authors' response

Thank you for highlighting this error, we have now changed 'delivered' to 'commenced' in the abstract.

Comment 2

Screening: "full-text articles were screened independently by two reviewers (VW and ADB or JD) and reasons for exclusion recorded." Were results of screening compared, if so how, and were there any discrepancies?

Authors' response

The result of each reviewer's screening was recorded in an Excel Spreadsheet. The screening results were then compared and any discrepancies in decisions regarding eligibility were highlighted. The two reviewers then met to review the full text papers of the studies and discuss these discrepancies. Using this method all initial discrepancies were resolved. We have now added a brief explanation of our approach into the manuscript on page 6: "Results of screening were compared, and any discrepancies were resolved through further review of the full text articles and discussion between reviewers".

Comment 3

Data extraction: define "serious adverse event" and perhaps indicate how this compares to a non-serious adverse event. Appendix 3 is noted – are these only 'serious' adverse events?

Authors' response

Thank you for this comment, this should actually have been 'adverse events', rather than 'serious adverse events'. As you point out, not all the adverse events reported in Appendix 2 are serious. We have now amended any reference of 'serious adverse event' to 'adverse event' in the manuscript. We have included a definition of 'adverse event' on page 6 of the manuscript: "any untoward medical occurrence in a clinical study participant regardless of the causal relationship with the study treatment".

Comment 4

Figure 1: why exclude interventions that started before hospital discharge and exclude interventions that started > 3 months after the operation (accepting that interventions will need to be complete to 12 months)?

Authors' response

This is an important point, and the Editor also requested further clarification about our rationale for only including post-discharge interventions that commenced in the first three months after surgery. Our response to this, and the actions we have taken to clarify this in the manuscript, are provided in our response to the Editor's first comment.

Comment 5

Results: The meta-analysis included 6 studies. Five studies compared the 'intervention' to 'usual physiotherapy care'. (Table 4 indicates 'usual physiotherapy care', but this is not defined or described; doing so would assist the reader to understand the comparison intervention). Regarding Figure 2, none of these five studies found a difference between the groups. The remaining study compared exercises with no physio/advice to stay active – this study found a difference between the groups. The conclusion "that post-discharge physiotherapy interventions appear not [to] be effective at reducing the severity of chronic pain after TKR" is not supported by the results, given that 5 of 6 studies compared two types of physio. A more reasonable conclusion from these five studies is that there is no evidence that one type of physiotherapy produces different effects on pain 12 months following the operation compared to a different type of physio/exercise. The conclusion from the remaining study, is that exercise is more effective than advice to stay active. Given the fundamental difference between Monticone's study and the remaining five, the study's inclusion in the meta-analysis with the remaining five studies is questionable.

Authors' response

That you for raising these important methodological points, we have made extensive edits to the methods and discussion sections of the manuscript to address these issues. Firstly, we have now included details of usual care in Table 4. Based on your helpful suggestions, and those from Reviewer 2, we now feel that it would be inappropriate to pool the data in meta-analysis and therefore we have converted to a fully narrative synthesis and removed Figure 2. Our original intention was to compare the effectiveness of an intervention with minimal care. However, usual care was highly varied across the studies included in the review, and it was rare for patients in the control group to receive no care at all. Therefore, we are omitting our pooling of data in the meta-analysis as we now believe that this may be misleading because of the variation in the treatment received in the usual care groups. Converting to a narrative review does not change our findings, and there still remains only one study which evaluated an intervention that was found to have a beneficial impact on chronic pain after TKR. We have changed our interpretation of the findings to highlight that our conclusion is that there is no evidence that one type of physiotherapy intervention is more effective than another at reducing the severity of chronic pain after TKR.

Comment 6

Discussion: "Only 20% of patients will develop chronic pain after TKR [2] and delivering physiotherapy to all patients may reduce the ability to detect clinical benefit in terms of pain severity. This suggests a need to identify patients at high risk of chronic pain and provide these patients with intensive and

comprehensive interventions that have been specifically designed to reduce their risk of developing chronic pain.” Please provide evidence for this suggestion.

Authors’ response

We have rewritten this to highlight that this is a recommendation for future research, rather than as an evidence-based statement. It now reads: “Only 20% of patients will develop chronic pain after TKR (Beswick et al. 2012) and delivering physiotherapy to all patients may reduce the ability to detect clinical benefit in terms of pain severity. In the future, interventions might be more effective if they include processes to identify patients at high risk of chronic pain and provide these patients with intensive and comprehensive interventions that have been specifically designed to reduce their risk of developing chronic pain” (page 15).

REVIEWER 2: DAVID HAMILTON

Many thanks for inviting me to review this paper which evaluates the effectiveness of post-discharge TKA interventions in reducing the severity of chronic pain through systematic review. I think this is a nice study that has been performed technically well, as would be expected from this group. I have no concerns with the analysis. I do have a couple of general concerns though. These may simply relate to the wording and discussion as opposed to any flaw in methodology.

Comment 1

The study aim was to evaluate the effectiveness of interventions at reducing chronic pain severity, but I don't see much reference to pain severity in the manuscript. I may have misread this, but the entry criteria seems to be a report of pain outcomes by intervention group - and chronic pain defined as pain levels that are equivalent to (or greater than) baseline values? Are you measuring pain severity here, or rather the number of patients with chronic pain in the cohort?

Authors’ response

Thank you for raising this point, which we hope we have now clarified in the manuscript. Our primary outcome in this review was pain severity at 12 months or longer after TKR. All studies including in this review reported the effects of the intervention on continuous pain severity scores, and therefore we report pain severity scores in Appendix 4, rather than the number of patients with chronic pain. We have clarified that we were evaluating pain severity throughout the manuscript.

Comment 2

Following on from that, the definition of chronic pain is a little vague here – as there seems to be no note or threshold as to intensity? Is this patients with high levels of chronic pain at 12 months – that I would want to refer for chronic pain interventions and/or analgesia review, or is that patients that have not changed their reported pain intensity from pre-op to post-op. If it is the latter, we need to be a little cautious as to the interpretation of the term ‘chronic pain’.

Authors’ response

Again, this is another important point. There is little guidance around what severity of pain counts as chronic pain. The most recent definition of chronic post-surgical pain suggests that it should be ‘should be of at least 3–6 months’ duration and significantly affect health-related quality of life’ (Werner et al, 2014). However, there is little guidance around how to quantify a ‘significant effect on health-related quality of life’, and it would not be possible to elucidate this from the measures of pain severity used in the trials included in this review. Also the majority of trials only report average pain severity scores, and therefore it is not possible to determine the severity of individual patients’ pain. Therefore, our definition of chronic pain is necessarily broad to evaluate whether post-discharge interventions were effective at reducing the severity of pain at 12 months or longer after TKR. This is an approach that we have taken in previous systematic reviews (Wylde et al 2017a Wylde et al 2017b, Beswick et al, 2015) Werner MU, Kongsgaard UE. I. Defining persistent post-surgical pain: is an update required? Br J Anaesth 2014;113(1):1-4.

Wylde V, Beswick AD, Dennis J, Gooberman-Hill R. Post-operative patient-related risk factors for chronic pain after total knee replacement: a systematic review. *BMJ Open*, 2017 Nov 3;7(11):e018105
Wylde V, Dennis J, Beswick A, Bruce J, Eccleston C, Howells N, Peters T, Gooberman-Hill R. Systematic review of the management of chronic pain after surgery. *British Journal of Surgery*, 2017; 104(10):1293-1306.

Beswick A, Wylde V, Gooberman-Hill R. Interventions for the prediction and management of chronic postsurgical pain after total knee replacement: systematic review of randomised controlled trials. *BMJ Open*, 2015, 5(5):e007387.

Comment 3

I wonder as to the rehabilitation goals of the included studies. Specifically, I wonder whether the included studies were aiming to reduce levels of chronic pain or whether they were generally aiming to enhance outcomes. Though clearly similar intentions at face value, I would agree that these are not the same thing. It is interesting that the study that did seem to influence pain outcomes was the one that addressed kinesiophobia. The others that did not influence pain scores seem to be generic exercise based interventions – I would not necessarily expect exercise interventions to address pain. On balance, I think that what has been looked at here are the randomised interventions delivered in the post-discharge period that aim to improve outcomes following TKA (and that record pain levels). I think the authors show a lack of efficacy as to these in reducing pain report at 12 months in relation to the internal control of the study.

I agree with the conclusions that uniformly applied intervention don't seem to be particularly effective at influencing outcomes – but I'm not entirely sure the authors can comment on chronic pain severity? Perhaps I'm mis-reading things, but I think there needs to be a bit more context given as to what is actually being measured here.

Authors' response

Thank you for these helpful comments, we have extensively revised the results and discussion sections of the manuscript in response to these comments and those provided by Reviewer 1, on which there is some overlap. We have expanded the Table in Appendix 4 to include the aim and primary outcome of each trial. In the manuscript on page 9, we have highlighted that of the 13 trials which specified a primary outcome, this was function in eight trials; a composite of pain and function in four trials, and pain in one only trial. We have also added a paragraph to the discussion pages 14-15 to address this issue.

As outlined in our response to Reviewer 1, we have removed the meta-analysis due to the variation in the treatment provided in the control group, and we have converted to a narrative synthesis. We have also extensively changed our interpretation of the findings. Our main conclusion is now that no specific type of post-discharge physiotherapy interventions appears to be effective than another at reducing the severity of chronic pain after TKR, while acknowledging that the primary aim of these trials was predominately to improve function.

The review also highlights the need for more evaluations of non-physiotherapy interventions, and trials with pain severity as the primary outcome. Our discussion about uniformly applied interventions is now written as directions for future research, rather than as an explanation for our findings.

VERSION 2 – REVIEW

REVIEWER	Stephen Gill Barwon Centre for Orthopaedic Research and Education (B-CORE), Australia
REVIEW RETURNED	01-Jan-2018

GENERAL COMMENTS	Thank you for updating the manuscript. It is disappointing that a meta-analysis was not deemed possible, but given the description of heterogeneity, this seems reasonable. I think the conclusions are now more defensible. Although the between group comparisons did not show differences (and these are the most important comparisons), it would be interesting to indicate whether participants in both groups changed (improved, got worse). For example, if both groups improved, there could be an effect due to physio, which would provide a hypothesis for further research such as comparing physio to usual care (as one study already suggests an effect).
---

REVIEWER	David Hamilton University of Edinburgh, UK
REVIEW RETURNED	13-Dec-2017

GENERAL COMMENTS	I think the authors have satisfactorily cleared up the questions raised in the original reviews. I have no qualms about recommending its acceptance. This is a very useful paper that I expect to quote.
--

VERSION 2 – AUTHOR RESPONSE

Comment from Reviewer 1

Thank you for updating the manuscript. It is disappointing that a meta-analysis was not deemed possible, but given the description of heterogeneity, this seems reasonable. I think the conclusions are now more defensible. Although the between group comparisons did not show differences (and these are the most important comparisons), it would be interesting to indicate whether participants in both groups changed (improved, got worse). For example, if both groups improved, there could be an effect due to physio, which would provide a hypothesis for further research such as comparing physio to usual care (as one study already suggests an effect).

Author response

Thank you again for your previous comments, we agree that the manuscript is now much stronger because of the improvements recommended by yourself and Dr Hamilton.

Thank you also for your current comment regarding change scores. We agree that it would be useful to present these for each trial for transparency, and we have added these additional data into the trial results summary in Appendix 3. Both groups improved in all the studies, which is to be expected, as the majority of studies conducted the baseline assessment prior to surgery. However, because the baseline assessment was prior to surgery it is not possible to say whether the improvement is due to the surgery or the physiotherapy interventions. There were four studies that conducted a post-operative baseline assessment but these were still early in the recovery trajectory (3-4 weeks, 6 weeks, and 2 months) and therefore improvements in pain would be expected as pain outcomes can continue to improve up to 12 months after surgery. Therefore it is difficult to draw conclusions from this data, but we agree that it is useful to present this data in Appendix 3.